# Using fatherhood to engage men in HIV services via maternal, neonatal and child health entry points in South Africa

Cathrine Chinyandura [1,2]*, Natasha Davies[1], Fezile Buthelezi[1], Anele Jiyane[1], Kate Rees[1,3]

1 Anova Health Institute, Johannesburg, South Africa, 2 Department of Sociology, University of the Witwatersrand, Johannesburg, South Africa, 3 Department of Community Health, School of Public Health, University of the Witwatersrand, Johannesburg, South Africa

* chinyandura@gmail.com, chinyandura@anovahealth.co.za

## Abstract

### Introduction

In South Africa, uptake of HIV services remains lower amongst men compared to women, resulting in poorer clinical outcomes. Several factors contribute to this situation, including stigma, confidentiality concerns, inconvenient clinic operating hours, fear of an HIV-positive test result, and long-waiting times. Additionally, women living with HIV are frequently identified whilst accessing other routine services, particularly antenatal and well-baby care. Novel approaches and strategies are needed to increase men's routine utilization of health services. For many men, fatherhood is an important part of being a man. Maternal, neonatal and child health services (MNCH) present an opportunity to improve male engagement with routine health services and subsequent uptake of integrated HIV care. However, men's involvement in MNCH services remains low. This study explored the concept of fatherhood and factors influencing men's involvement in MNCH services.

### Methods

This was an exploratory, qualitative study. Three focus group discussions (FGDs), involving 33 male participants, were conducted with men living in communities across Johannesburg. Men were recruited by male peer counsellors, employed by Anova Health Institute under the men's health programme. Data was collected between May and July 2021. Authors had no access to information that identify individual participants during or after data collection. Data were transcribed inductively and analyzed thematically using NVivo software.

### Results

The study found that male participants were eager to be involved in MNCH services. They valued fatherhood and were making concerted efforts to be involved fathers. However, multiple factors influenced men's involvement in MNCH services. Barriers included sociocultural norms, employment commitments, boredom and disengagement while waiting for services, negative staff attitudes and long waiting times. Participants identified multiple facilitators

**Data Availability Statement:** We have provided transcripts excerpts attached as Supporting information.

**Funding:** This research has been supported by the President's Emergency Plan for AIDS Relief (PEPFAR) through the United States Agency for International Development (USAID). The funders had no role in study design, data collection and analysis, decision to publish, or preparation of the manuscript.

**Competing interests:** The authors have declared that no competing interests exist.

that would encourage their attendance at MNCH services including positive staff attitudes, quick service, active engagement, positive affirmations by health care workers and the visibility of male health workers' in MNCH spaces.

## Conclusions

The study highlights that men strongly desire to be involved fathers and included in MNCH services. HIV programmes should support this and harness it to actively engage men in HIV services. However, to encourage greater male involvement in MNCH, socio-economic and healthcare system related factors need to be addressed when designing strategies that create more inclusive, family-orientated, male-friendly, and integrated MNCH services.

## Introduction

Major gender gaps exist in the uptake of HIV services despite increased service availability across South Africa. Uptake of HIV testing and treatment remains lower amongst males compared to females, resulting in poorer clinical outcomes, and presenting a significant barrier to achieving the UNAIDS 95-95-95 targets [1]. Achieving these targets calls for inclusive people-centered approaches towards the HIV response. Engaging men as agents of change is a critical component of HIV prevention and care [2, 3].

Several factors contribute to lower uptake of HIV testing, prevention, and treatment services among men, including stigma, confidentiality concerns, inconvenient clinic operating hours, fear of an HIV-positive test result, and long-waiting times [4–6]. Moreover, women have more regular contact with formal health services than men do, either for their own health (for example for contraception or antenatal services) or for their children [7]. This increases their likelihood of accessing other HIV services compared to men who have less regular contact. Novel approaches and strategies are needed to increase men's routine utilization of health services. Maternal, neonatal and child health services (MNCH) present an opportunity to improve male engagement with routine health services and subsequent uptake of integrated HIV care.

Men play a crucial role in MNCH as their influence on decision making and provision of material support can impact their partner's health care seeking behavior and access to services [8, 9]. In the context of HIV/AIDS, evidence indicates that male partners influence their partners' utilization of HIV prevention services and adherence to antiretroviral treatment [10–13]. Men's involvement in MNCH services is not only beneficial for men's partners and children but is also beneficial to their own health. Through MNCH services, men can access HIV services and other male-related services. In addition, MNCH services are strategically positioned to affirm responsible roles men can play in their families and help them realize how self-care can positively influence their role as fathers [14, 15].

Fatherhood is an important phase in a man's life and is often central to their identity. In most societies, fatherhood remains a valued social role [16]. Fatherhood has been found to have a positive impact on men's sense of responsibility, self-worth, mental/emotional health, and confidence [17]. Research studies with fathers in various settings revealed that fatherhood brings significant changes to men's lives, men reported making multiple lifestyle changes when they became fathers: new priorities, a change in masculinity, reorientation of time, increased responsibility, a shift in values, altered sense of purpose, increased problem solving abilities, and improved emotional regulation [18, 19]. Fathers also reported becoming more

mature, confident, secure, and empathic [18]. Paternal commitments have also been linked to improved men's health and religious participation [20]. Research indicates that men strongly desire to be involved in their children's lives and are unhappy when they are marginally or not involved [21]. MNCH is therefore a strategic option to harness men's desire to be involved in their children's lives through active involvement in MNCH services.

Considerable efforts have been made to involve men in MNCH services, however the focus has mainly been on how their involvement can improve MNCH outcomes [11, 22–25]. Extensive research has been conducted on barriers and facilitators to men's involvement in MNCH services, and efforts have been made to address some of the barriers, however, men's involvement in MNCH services remains low [26]. New approaches and perspectives of increasing men's involvement in MNCH services are therefore needed. The links between male engagement in MNCH and its benefits to their own health and well-being are not well documented. In addition, the positive role that fatherhood can play in improving men's engagement in MNCH services is not well explored. Leveraging on men's roles as fathers, and their interest in building healthy families, may have the potential to motivate them to utilize MNCH and, subsequently, HIV services and this needs to be further researched as a possible approach to close the HIV gender gap.

This study explored the concept of fatherhood and how fatherhood and other factors influence men's involvement in MNCH services, with the aim of contributing to the design of programmes to encourage male involvement in both MNCH and HIV services.

## Methods

### Study setting

The study was conducted in Johannesburg District, Gauteng province, South Africa from May-July 2021. Johannesburg has a population of approximately 4, 8 million (males 2 419 151; females 2 384 111) [27]. This population is spread over 7 regions, with distinct socio-economic characteristics. FGDs were held in various community sites including a shebeen (informal licensed drinking place in the community) and clinic meeting rooms. The sites were selected to ensure representation of sub-district management clusters and recruitment of diverse group of men. The sites were decided upon in consultation between participants and male peer counsellors. The sites were sometimes used by male peer counsellors to conduct community dialogues, educational awareness sessions and during community mobilization campaigns.

### Study design and participants

A qualitative exploratory study using FGDs was undertaken. We conducted three (3) FGDs, with a total of 33 men selected from various geographical regions in Johannesburg. One group was composed of male peer counsellors working for Anova Health Institute in the Health4Men programme. The male peer counsellors are based at clinics spread across Johannesburg District and have the responsibility of community mobilization and providing supportive client-centered services at the clinics to men living with HIV. Two (2) groups were constituted of men drawn from local communities. Some of the men had participated in community Health4Men Programme interventions while some had never participated in the interventions. The FGDs comprised of 6–12 participants. The FGDs lasted between 90–180 minutes.

Eligible participants were men who were above the age of 18 years old regardless of them having biological or non-biological children, their nationality, ethnicity, race, religion, social status, and educational level. The exclusion criterion were ages less than 18 years and inability to speak in English and local languages. Convenience sampling was used to recruit participants in the study. Male peer counsellors were recruited in the study through an invitation circulated

to them by the Health4Men programme lead. Men from local communities were recruited through male peer counselors who mobilized the men to participate in the study during their community mobilization activities.

FGDs were conducted by research staff in English and local languages (isiZulu, isiXhosa and sePedi) and audiotaped for subsequent transcription, with simultaneous translation into English. FGDs facilitated gathering of collective views and opinions regarding fatherhood and men's involvement in MNCH services. The method also allowed the researchers flexibility, probing for further discussion of important and relevant issues that arose during the discussion. The FGD guide covered the following topics: understanding fatherhood, fathering experiences, benefits and challenges of fatherhood, self-care, facilitators, and barriers to utilization of MNCH services and preferred incentives to encourage accessing care. The FGD guide was used to guide the discussions but the FGDs adopted a loose structure to allow participants to share their experiences, perceptions, views, and insights openly and freely with minimal interference.

## Ethical approval

All methods were carried out in accordance with relevant ethical guidelines and regulations. Written informed consent was obtained from all participants before conducting the FGDs. Participants were given identifiers to ensure privacy and confidentiality. Ethical approval was obtained from the Human Sciences Research Council (HSRC) REC 3/22/08/18 and the Johannesburg District Research Committee.

## Data analysis

The FGDs were audio recorded, transcribed, translated, and imported into NVivo 12 qualitative data analysis software. The software was used for coding, categorization, and identification of themes. Thematic analysis was used to analyze the qualitative data. Two research facilitators fluent in local languages transcribed the data, and the generated transcripts were independently reviewed line by line before coding by the team lead who supervised the data collection process. Data were analyzed inductively by organizing it into categories based on themes, concepts, or similar features. A total of 3 team members divided into 2 independent teams analyzed the data. The first team was composed of team members who had been involved in data collection. Each member of these team members independently generated initial codes through open coding. Each member independently coded to bring individual perspectives which could be linked to their earlier engagement in data collection. This involved repetitively reading through the data and assigning codes. This was an iterative process as the initially assigned codes were subject to change as coding progressed. The created codes were then categorized into sub-categories, with each sub-category consisting of similar codes, in terms of topic or general concept. The first two team members then merged their files and re-examined the resulting codes and categories to develop final codes, categories, and themes. The re-examination allowed for a discussion on the rationale of the codes, categories, and themes created by each member.

The second team was composed of only one team member who independently reviewed the codes, categories, and themes. The team member had not been involved in data collection. The second team member reviewed the coding of the merged file from the first two members. The third team member was included to provide a different perspective as they were not part of the data collection process unlike the first two team members. This entailed re-naming, re-coding, and re-categorizing the data as needed. The FGD guide was used to facilitate reflexivity and to ensure the consistency of the processes of coding. The two teams then reviewed the

merged and reviewed second dataset to refine and reach consensus on the codes, categories, and themes. The review by both teams also explored the emerging associations and inconsistencies in the categories and themes.

The final themes were created through interrogating similarities, differences and connections of created categories. A matching and comparison examination was done on the created categories. Careful consideration was made to ensure that the examination does not outstrip the data of its context. As we matched and compared the categories, we noted emerging patterns and underlying themes through construction of the semantic and latent meanings in the categories. These themes were then used to construct the final narrative.

# Results

## Demographic characteristics of participants

Table 1 summarizes the demographic characteristics of the participants.

The ages of the participants ranged from 25–74 years. The majority of participants (12) were in the age group of 35–44 years. The majority of participants (22) were employed while nine (9) of the participants were unemployed. All the participants identified as Black African. Twelve (12) of the participants reported that they had 1–2 biological children, another 12 participants reported that they had 2–4 biological children and only one participant reported not having biological children. The majority of participants (16) reported having non-biological children. All participants (31) considered themselves to be involved fathers. The majority of participants (24) reported residing with their children. Participants were of mixed ethnic backgrounds, including Zulu, Xhosa, Sepedi, Venda and Tsonga.

## Fatherhood

The fatherhood construct disintegrated into sub-themes to bring a holistic understanding.

The fatherhood theme explored the attributes of a father, changes to fatherhood in South Africa, rewards and challenges of fatherhood, factors influencing the level of involvement of fathers and support needed by fathers to fulfill their roles.

**Attributes of a father.** When asked about the characteristics of a good father, participants highlighted that a good father is available, responsible, a teacher, provider, protector, and role model. Participants stressed that fatherhood was more than material provision, they regarded a father's involvement in his children's lives to be an important aspect which could not be replaced by material things.

*"Being a father means being there for your child in every way not only financially, but you have to play your role because as fathers when we give money to our children, we think we are there for our children".*

*(FGD 1, P6)*

*Changes in fatherhood.* All participants believed that fatherhood had changed over time in South Africa. Some of the changes were for the better while some were perceived to be harmful. Participants pointed out that the current generation of fathers was making concerted efforts to be involved in their children's lives compared to past generations. Better employment opportunities and educational levels were reported to be some of the factors which encouraged fathers to take more active roles in raising their children.

**Table 1. Profile of the participants.**

| Variable | n |
|---|---|
| **Age group** | |
| 25–34 | 7 |
| 35–44 | 12 |
| 45–54 | 10 |
| 55–64 | 1 |
| 65–74 | 1 |
| Missing responses | 2 |
| **Employment status** | |
| Employed | 22 |
| Unemployed | 9 |
| Missing responses | 2 |
| **Race** | |
| Black African | 31 |
| Missing responses | 2 |
| **Number of biological children** | |
| 0 | 1 |
| 1–2 | 12 |
| 3–4 | 12 |
| 5–6 | 4 |
| 7–8 | 2 |
| Missing responses | 2 |
| **Number of non-biological children** | |
| 0 | 9 |
| 1–2 | 16 |
| 3–4 | 5 |
| 5–6 | 1 |
| Missing responses | 2 |
| **Considers himself to be involved father** | |
| Yes | 31 |
| No | 0 |
| Missing responses | 2 |
| **Currently resides with any child** | |
| Yes | 24 |
| No | 7 |
| Missing responses | 2 |

*"Men are stepping up and they are taking responsibility to be there for their children. You will find that it is us men who are fighting to be there for our children unlike before".*

(FGD 1, P4)

**Level of involvement.** Participants explained that fathers' level and quality of involvement depended on several socio-economic factors. All participants stressed that poor relations between a father and the children's mother was a key contributing factor to poor involvement with their children. Participants explained that mothers of their children often expected fathers to "buy" access to their children via contributions of formal or informal child support.

*"The one thing that hinders a man to be present in the life of the kids is the mother of the child, women take out their revenge by using the kids to pay back on us".*

(FGD 2, P5)

Strained relationships with in-laws and failure to fulfill cultural obligations such as payment of *lobola* (bride wealth) and/or "damages" (payment for fathering a child without being married to the mother) were reported as other key factors impacting the level of fathers' involvement in a child's life.

*"We earlier spoke about having children when we were still young, and then came the issue of paying for damages. Where am I going to get the money for paying damages whilst I am still young, and I am still at school. I still do things for the child but there is still that thing that I need to pay for the damages and the child is using the mother's surname. The child cannot change to mine until I have paid the damages".*

(FGD 1, P1)

*"I would put this on the parents of the girl. The parents don't appreciate what you do for the child. It won't be enough; they would compare your support with others".*

(FGD 1, P3)

For those who were employed, work schedules and demands were identified as major constraints to paternal involvement in childcare. Participants highlighted that work demands limited their involvement in their children's lives and family life in general. However, conversely, some participants added that being unemployed was also a major limitation in fulfilling their roles, especially that of a financial provider.

Participants underscored that social norms on gender roles discouraged men from taking an active role in childcare as it was considered a female domain.

*"What discourages on the other side is the society, society standards discourage us from supporting our women and children. For instance, if I may ask all of you here how many of you buy [sanitary] pads for your girlfriend, and do you even know the brands? Because if I walk with pads in the street, people will say I am not a man enough. Society standards discourage us from supporting women and children that way".*

*(FGD 2, P5)*

Despite all these constraints, participants reported that they strived to have high-quality relationships with their children. Participants found fatherhood to be rewarding, which motivated them to be better fathers. All participants reported being elated when their children were born and shared memorable experiences of the time they spent with their children.

*"When I heard that I am going to be a father. I was excited because I felt that I am a man now and that my father's surname is growing".*

(FGD 3, P4)

Participants narrated how fatherhood had made their lives meaningful, promoted a sense of worth and improved their social status.

*"I would have died if I didn't have kids because I didn't have a reason to live. I was doing things the way I want".*

(FGD 3, P4)

It was clear from the discussions that participants valued fatherhood and endeavored to establish strong and positive father-child relationships. All participants stressed that most men strongly desired to be more involved and play an active role in their families' lives.

**Support needed.** Participants acknowledged that they required different types of support to be involved as fathers. Participants reported that most men lacked knowledge and information on their roles and contributions towards healthy child development. In addition, participants pointed out that most men in South Africa were raised in dysfunctional families characterized by abuse, neglect, and absent fathers. Consequently, they were emotionally broken and lacked skills and guidance on fathers' roles and responsibilities.

*"Most fathers need a lot of engagement, why am I saying this? Because we are from broken societies you understand, through the way we grew up, we grew up in the environment where our mothers were beaten up and we thought its normal, we grew up in an environment whereby alcohol comes first and it's a norm. So, we were broken from the beginning when we were kids, hence we are broken even now, hence breaking others, if we were raised with love, it will be easy to communicate our emotions to our partners to say, please love me. Connection is important and if we had that we were not going to run away from our kids".*

(FGD 1, P1)

Participants were interested in forums where they could learn and exchange with other men. Participants perceived that peer support groups or forums provided safe and private spaces where they could freely share their struggles and gain information and skills on parenting. These supportive social networks were thought to provide a sense of community and opportunities for both young and old fathers to learn through shared experiences.

*"What I think the kind of support that we need is having leadership conversations and workshops where we will have fathers who will groom young men to be good fathers in society because money is nothing without Ubuntu (humanity towards others)".*

(FGD 2, P5)

### Barriers and facilitators to male involvement in MNCH services

The study explored factors that encourage and discourage men from accessing MNCH services. The data revealed that several factors at different levels influence men's involvement in MNCH services. These factors fall into two categories: socio-economic and health-system related.

**Socio-economic barriers.** *Employment/unemployment.* Employment commitments were identified as a key barrier to men's utilization of MNCH services. Most participants expressed willingness to support their partners and children but were constrained by work demands and routines. Participants explained that work schedules made it difficult to attend MNCH services as they were incompatible with clinic operating hours. Participants stated that men felt compelled to primarily direct their energy and time towards work as they are traditional family providers.

*"I would say employment, time to do all those things, finding a balance is quite challenging".*

*(FGD 1, P4)*

*Cultural and gender norms.* Participants highlighted that socio-cultural norms on gender roles deterred men from actively engaging in childcare and MNCH services. Participants in all groups explained that nurturing and caring for children were considered "women's work" and men were directly or indirectly discouraged to take an active role in such affairs. Participants reported these social norms were also mirrored in MNCH services where HCWs' attitudes were not supportive of fathers who were involved in childcare and the clinic spaces being female dominated.

*"Nurses tend to make mean comments, why they ask us about the mother of the child when I bring the child to the clinic? It's as if to say this is a place for women and not men"*

*(FGD 1, P3)*

**Health system related barriers.** The health system barriers described were related to accessibility challenges due to the way health services are organized and delivered and the overall environment of the health facility. The following health system factors were identified: negative health worker attitudes, long waiting times, boredom and disengagement at health facilities, and female dominated spaces.

*Negative staff attitudes.* Participants expressed that they were disheartened by the negative attitudes and behaviors of health care workers (HCWs) towards them, which they said discouraged them from accessing MNCH services. Participants raised concerns that HCWs, particularly nurses, were unfriendly and addressed them rudely, with no respect, compassion, or empathy. Participants felt "degraded or demeaned" by HCWs, which often impeded them from accessing health services. Some of the participants recounted how HCWs had ill-treated them when they visited the health facilities.

*"Nurses asked me why are you bringing the child, where is the mother? This thing limits men. It is sad when we come to the clinic and then you get disrespected by a woman that thing is not nice. This thing ends up making us not want to come to the clinic because you will think of the disrespect that you will get from the nurses but when you look at the doctors, they respect a patient a lot, but the nurses are disrespectful".*

*(FGD 3, P4)*

*Long waiting times.* Our results showed that waiting time affected men's utilization of healthcare services and their satisfaction with care they received. Long waiting times were reported as one of the major hindrances to men utilizing MNCH services and health services in general. All groups universally felt that the long waiting times for services was incompatible with men's demanding schedules, hence their reluctance to access services.

*"The thing is you end up staying 4–5 hours outside so that discourages us a lot, if it is just me and the baby its better because I am the older person who is with the baby, but if it's the 3 of us (the baby, mother and I) then it becomes hard".*

*(FGD1, P5)*

*Boredom and disengagement at health facilities.* Lack of active engagement and stimulation while waiting for services was mentioned as an obstacle to men's utilization of MNCH services. Participants observed that the environment of most health facilities was monotonous and lacked stimulation resulting in boredom. Participants added that the boredom was worsened when HCWs instructed them to wait outside when they accompanied their partners and children to the health facility.

*"You come to the clinic, and you are sitting there and doing nothing I think that is what makes men not want to come to the clinic. I think having something to keep men engaged will encourage them to come to the clinic even more".*

*(FGD 3, P6)*

*Female-dominated spaces.* Participants described MNCH services as not male-friendly due to the dominance of women. The heavy presence of women, either as users or service providers in MNCH spaces was perceived as a deterrent to men. Participants expressed that men often felt alienated and uncomfortable in the female dominated spaces.

**Proposed solutions.** This section outlines participants' suggestions on ways to facilitate or encourage male involvement in MNCH services.

*Positive staff attitudes.* The quality of interactions between service providers and patients was reported to have an influence on men's utilization of health services in general. Participants noted that men were likely to access MNCH services if they received warm, favorable, and respectful treatment from HCWs. Participants emphasized that respect was a key element to men, and it should be demonstrated in the way HCWs communicate and interact with patients.

*"If nurses can give us attention and respect. Someone who will assist me as I come in, someone who is warmer in welcoming".*

*(FGD 1, P4)*

*Positive affirmations from HCWs.* Participants highlighted that men feel motivated and are likely to return for services, remaining engaged, when HCWs affirm that their behavior can positively impact their own, their partner's and their child's health. Participants explained that affirmations had the power to reinforce positive behavior and increase men's confidence. The more men were praised for their positive behavior, the more they were driven to continue the positive acts.

*"I can speak on my side. I can say encouraging us, talking to us. As you are a nurse when a father brings the child, encourage him and talk to him, have positive words for the father so that he gets encouraged".*

*(FGD 1, P7)*

*Quick service.* Participants in all groups underlined that they preferred accessing services at facilities where they did not have to wait for long periods. The discussions revealed that participants' demand for timely service were related to work-related demands which often left them with limited time for other commitments.

*"I am saying time, if you say 2 hours, it must be 2 hours it shouldn't be more. Most men I know and work with have the mentality that women spend so much time in the clinic on purpose. They think their time is not valued; women do not take time seriously".*

*(FGD 3, P2)*

*Flexible hours.* Participants expressed that lack of extended operating hours in most public health facilities served as a barrier to men accessing MNCH services. Participants suggested that health facilities should have flexible operating hours to accommodate users, especially men who due to competing priorities, mainly work-related responsibilities, could not access services during normal working hours and days.

*"If these clinics can open hours which suit us, like after 5 and weekends then as fathers we would probably show up".*

*(FGD 2, P8)*

*Active engagement.* Participants emphasized that men prefer to be actively engaged in various ways while waiting for services to avoid boredom. Participants felt that there were many useful ways health facilities could actively engage men during the long waiting times.

*"Sharing of experiences, if there might be a person that sits outside sharing their experiences, and interact with me, they can share something that can encourage you as the father. That way I cannot go back angry and bored, and it encouraged me to be a better father".*

(FGD 3, P5)

*Male health care workers (HCWs).* Visibility of male HCWs in MNCH services was suggested as a strategy which would put men at ease, make the services male-friendly, and ultimately encourage their involvement.

*"As men being assisted by women is another issue, so we need more male clinicians; someone who will understand what you are talking about because they have also maybe gone through the same thing".*

*(FGD 2, P4)*

## Discussion

Our study has shown that men value the fatherhood role, with it being a central component of their identity. All participants appreciated their role as fathers and expressed a strong desire to be involved in their children's lives. Participants also narrated the concerted efforts they were making to be involved fathers. MNCH services can seize the opportunity of men's desire to be involved fathers, and harness that to increase service utilization, and affirm that taking care of their own health would allow them to provide and care for their children. MNCH services are well positioned to build on men's roles as fathers, develop their capacity as fathers, provide knowledge, skills, and confidence to be more involved with their children. However, the challenge is that the HCWs offering these services lack the skills to do so. Although men are already involved in different ways in MNCH, the involvement is mostly "facilitative", for example, escorting their partners to the health facility and providing financial resources for

transport costs and any medical expenses. MNCH services need to enable men to move beyond the "facilitative" role towards active involvement.

Existing literature indicates that fear of HIV testing is a primary barrier to men's utilization of HIV services [5, 28, 29]. Moreover, men may be afraid to test with their partners due to fear that a positive HIV test could expose hidden infidelity, be blamed for being responsible for bringing HIV into the relationship, resulting in marital or relationship tension. Although men in our FGDs did not mention fear of HIV testing as a barrier to involvement in MNCH services, anecdotally, women participating in our MNCH programmes highlighted that their partners were reluctant to accompany them to MNCH due to fear of HIV testing. Even though MNCH services are a strategic entry point for HIV services for men, the efforts to get men involved in these services should be thoughtful not to be identified and perceived by men as a strategy to lure them for HIV testing and linkage to HIV services. Such perceptions can adversely impact on men's utilization of MNCH services. HCWs should therefore be sensitive to this when providing services. HCWs should rather focus on creating rapport and educating men on the benefits of self-care on fatherhood and manhood. The rapport and awareness would ease men's fears and encourage them to test for HIV and utilize integrated HIV services as they build up trust in the health services they are accessing. This is also applicable to HIV services; careful messaging is needed for men to understand the personal benefits of utilizing the services. It is important to help men realize that their involvement in HIV services allows them to regain their health, restore masculinity compromised by HIV, and provide for their families, particularly their children. These are key elements to men and can therefore be powerful motivators for engagement in care [6].

Barriers and facilitators to men's involvement in MNCH services need to be considered in the adaptation of services, and implementation of interventions to increase male engagement. Our findings show that men's involvement in MNCH is affected by socio-economic and healthcare system related factors which should be considered to create a more inclusive health services environment. Male engagement in economic activities to fulfill their role as family providers seem to be in tension with their desire to participate in MNCH services and family life in general. The findings indicate that employment schedules often infringe on family life, hindering men from being involved in MNCH services. The findings reveal the central role that work plays in men's identity and how it occupies much of their time. Timing and scheduling of services would, therefore, be critical to improving men's involvement in MNCH services and health services in general.

The findings clearly show that men's utilization of MNCH services is greatly influenced by socio-cultural norms. Socially constructed gender roles limit men's participation in MNCH services. The study findings are consistent with previous research studies which have highlighted that gender stereotyping of MNCH as women's primary activity discourage men from being involved MNCH services [11–13, 30]. In addition, MNCH services are predominantly occupied by females (female health workers and clients) making men feel alienated. Historically, MNCH services have been directed primarily at women which could have reinforced the perception of the services as women affairs. Engaging communities and HCWs should be considered to transform cultural norms that discourage men from accessing MNCH services. Working with community structures such as men forums and churches is crucial to transforming ingrained cultural norms which hinder men's involvement in MNCH services.

Participants noted that one of the key facilitators to involve men in MNCH services is increasing the visibility of male HCWs. It should be noted that the organizational structure of the healthcare system reflects gender imbalances in the healthcare professions. Men remain underrepresented in the nursing profession, a structural challenge which is unlikely to change soon. This is because healthcare professions like many other professions reflect

prevailing societal power and social relations. In most societies, nursing is still considered to be a female career because of gender stereotyping norms which consider caregiving to be a role for women. Even though socio-cultural norms are dynamic, it might take some time to witness a significant increase of men joining the nursing profession. Given this structural deficiency of the healthcare system, innovative approaches should be used to improve the visibility of male HCWs in MNCH services. This could involve enlisting male clinicians and non-clinicians (general support staff and peer supporters) to support MNCH services, for example through facilitating health talks during the waiting time for services and becoming male champions for MNCH services. The role of mentor fathers would also be useful in this regard. Mentor fathers would play a similar role as mentor mothers, a cadre enrolled in PMTCT programmes to provide psychosocial and other support to pregnant and post-partum women living with HIV. Mentor mothers have been found to enhance maternal-infant outcomes [31].

HCWs' conduct and attitudes towards patients play a central role in patients' utilization of services. A growing body of literature has shown the impact of staff attitudes on patient service satisfaction and utilization of services [32–35]. Negative staff attitudes are contrary to the ideals and ethical code of the health profession. Negative attitudes of HCWs in South Africa have been widely reported [35, 36], making it an area of concern requiring urgent redress to improve patient care and restore patient trust in the healthcare system. It is important to note that negative staff attitudes are due to multiple factors, including burnout, inadequate staff training, conformity to socio-cultural norms, work overload, lack of resources and lack of management support. These factors need to be considered when developing interventions to increase male involvement in MNCH services. Engaging HCWs is important to shift their attitudes around gender norms.

Our findings indicate that men are less likely to use health services when services lack privacy, are offered at inconvenient hours, and involve long waiting times. Enlisting men's involvement in MNCH services requires creating male-friendly and targeted interventions in terms of convenience, privacy, and positive patient–provider interactions. Our findings indicate failure of the health care system to adapt services to suit men's needs. Consequently, failure of the healthcare system to adapt accordingly has an impact on utilization of services and health outcomes. In broader terms, the rigidity and lack of responsiveness of healthcare systems to address male health needs has significant public health cost. In the absence of adaptation of HIV services to encourage men's uptake of the services, men will remain missing in the continuum. Consciously considering the needs and preferences of men has the potential to improve men's attitudes towards MNCH services. To improve accessibility of services, healthcare services should consider the needs and preferences of all their service users in line with a person-centered approach. Such considerations are likely to benefit men as well as other groups who face access challenges.

It was evident from the discussions that participants were unhappy with the current structure of services and how they were offered. Re-orientation of services towards a family-centered approach is likely to provide a favorable environment for men, encouraging them to attend MNCH services. The restructuring towards a family-centered approach might require allocation of additional resources to create more "family friendly" MNCH spaces. Making waiting areas and consultation rooms comfortable for families will require investments in resources, for example, additional spaces, provision of appropriate information, education, and communication materials. However, simple steps such as shifting the language used in MNCH spaces and by HCWs towards family-based language could have a positive impact on service utilization by men. Other initiatives include training of health workers on the delivery and promotion of family-centered services.

## Recommendations

More work remains to be done to improve men's engagement with MNCH and HIV services. Further efforts in both research and practice should focus on other strategies which can be used to motivate male engagement in MNCH and HIV services. Possible strategies which can be explored further include a) use of mobile applications; b) issuing a patient card to men who are expecting and/or with children attending Expanded Programme on Immunization (EPI) services; and c) collaboration with community men's forums. The patient card would serve as official documentation men could use to request time-off in their workplaces and a reminder of the appointment dates when they were expected at the health facility. Partnerships with existing community forums such as churches and other social and cultural organizations provide sustainable opportunities which are not costly. Interactive multimedia mobile applications are widely used, convenient and not bound in time and space making them a powerful tool to engage men in MNCH services. Family-centered approaches and principles provide exciting new directions to pursue in rethinking service delivery and have the potential to increase men's involvement in MNCH and HIV services. Implementation science approaches should be applied to understand how these interventions can be operationalized and, where they have been implemented, assess their effectiveness in improving men's engagement with MNCH services and subsequent uptake of HIV services.

## Limitations

The transferability of the study findings may be limited to men who are not interacting with health or community services. However, the findings offer valuable insights for strengthening MNCH services and integration with HIV services. The small number of FGDs conducted and limitations of the sampling approach may influence perspectives on fatherhood and engagement in MNCH services.

## Conclusions

Our study highlights that poor service utilization and male engagement remain persistent challenges in HIV programmes, resulting in poor outcomes for men and making it difficult to achieve the 95-95-95 targets. Male engagement in MNCH could become an important strategic entry point for promoting men's health and effectively reaching men with integrated HIV services. Men in South Africa are willing to be involved in MNCH services but there is a need to address the barriers which hinder their involvement. In addition, men strongly desire to be involved fathers and the healthcare system should harness this desire through active engagement of men in MNCH services. In broader terms, increasing men's utilization of health services will require the healthcare system to be more responsive to men's needs and to adapt and/or design services to meet those needs and rebuild men's trust in the healthcare services on offer.

## Supporting information

**S1 File.**
(PDF)

## Acknowledgments

We thank all the participants who participated in this research.

## Author Contributions

**Conceptualization:** Cathrine Chinyandura, Natasha Davies, Fezile Buthelezi, Anele Jiyane, Kate Rees.

**Data curation:** Cathrine Chinyandura, Fezile Buthelezi, Anele Jiyane, Kate Rees.

**Formal analysis:** Cathrine Chinyandura, Natasha Davies, Fezile Buthelezi, Anele Jiyane, Kate Rees.

**Investigation:** Cathrine Chinyandura, Fezile Buthelezi, Anele Jiyane, Kate Rees.

**Methodology:** Cathrine Chinyandura, Natasha Davies, Fezile Buthelezi, Anele Jiyane, Kate Rees.

**Project administration:** Kate Rees.

**Supervision:** Cathrine Chinyandura, Natasha Davies, Kate Rees.

**Validation:** Kate Rees.

**Writing – original draft:** Cathrine Chinyandura.

**Writing – review & editing:** Cathrine Chinyandura, Natasha Davies, Fezile Buthelezi, Anele Jiyane, Kate Rees.

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
