## [Decision Letter · Decision Letter 0]

22 Sep 2023

PONE-D-23-16181Using Fatherhood to engage men in HIV services via Maternal, Neonatal and Child Health entry points in South AfricaPLOS ONE

Dear Dr. Chinyandura,

Thank you for submitting your manuscript to PLOS ONE. After careful consideration, we feel that it has merit but does not fully meet PLOS ONE’s publication criteria as it currently stands. Therefore, we invite you to submit a revised version of the manuscript that addresses the points raised during the review process.

In addition to some of the comments raised by the reviewers, please also address the following in your revision:

1. In your Study Setting section, you mentioned that the study was conducted in South Africa from June-July 2021. In the section that follows, you mentioned that data was collected between May and June 2021. It is not clear why you have such inconsistency in your reporting of dates.

2. The inclusion and exclusion criteria for the FGD participants in not clearly spelt out. Please provide a clear inclusion criteria that was use to recruit participants.

3. How many FGDs were held?

4. In the data analysis section, you mentioned that the NVivo 12 software was used for "coding, categorization, and identification of themes". Further down in the same section, you also mentioned "Each member of the first team independently generated initial codes through open coding." Given that two independent individuals had done this exercise separately, what was the reasoning behind having the second team to re-do the same exercise including re-naming, re-coding, and re-categorizing?

5. By definition, a theme is a central unifying idea or the bigger issue that that emerges from your data. In the results section, you mentioned that fatherhood was one of the key themes identified. How does fatherhood emerge as a theme when in fact it is one of the characteristics used for inclusion into the study? Check the last line in the Results section. How were the themes identified?

6. Please make sure to follow all PLOS ONE guidelines when submitting your paper. For example, you did not include line numbers. 

We look forward to receiving your revised manuscript.

Kind regards,

Edward Chiyaka, Ph.D., MSc

Academic Editor

PLOS ONE

Journal Requirements:

Reviewers' comments:

Reviewer's Responses to Questions

**Comments to the Author**

1. Is the manuscript technically sound, and do the data support the conclusions?

Reviewer #1: Yes

Reviewer #2: Yes

2. Has the statistical analysis been performed appropriately and rigorously? 

Reviewer #1: N/A

Reviewer #2: N/A

3. Have the authors made all data underlying the findings in their manuscript fully available?

Reviewer #1: Yes

Reviewer #2: Yes

4. Is the manuscript presented in an intelligible fashion and written in standard English?

Reviewer #1: No

Reviewer #2: Yes

5. Review Comments to the Author

Reviewer #1: Title: Using Fatherhood to engage men in HIV services via Maternal, Neonatal and Child Health entry points in South Africa.

COMMENTS:

Introduction:

Paragraph 3: first, second and third lines say repeated message in the different ways.

Paragraph 4: third sentence too long looses meaning on the way.

Methods

Study setting

Paragraph 1: first sentence does not concur with what is in the abstract. When was this study conducted?

Second sentence: sounds incomplete.. “aged between 15+ years are estimated”

Whole paragraph 1: fatherhood and age “15+ years” sound incompatible??

Study design and participants

Paragraph 1: fifth sentence: when was the study conducted? “May and June 2021” there seems to be no consistency.

Sentence 8: does not align with the age 15 years participants.

Sentence 9: does not align with the age 15 years participants.

Ethical approval

Paragraph 1: “Written informed consent was obtained from all participants.” Participants underage (aged 15 under methodology above) cannot give consent.

Data analysis

Paragraph 1: sentences 5&6: Concord, revise.

Level of involvement

Concord: Double check whether the noun is singular or plural

Culture and gender norms

Concord: Double check whether the noun is singular or plural

Results

Paragraph 2: ages of 15+ and The age range of the participants was 23-67 years.’ Please check consistency.

My final Comment:

This study has vital inconsistencies in it:

1. Very important and legal issue of participants’ age to take part in the study has been overlooked when the study involved age 15+ participants. Towards the end the study changes from age 15+ to 23-67 years. There was no mention of steps taken to with regard to ages under 18. For minors younger than 18 years of age to participate in a research study, parental or guardian permission must be obtained. For minors a youth assent form is required.

2. the period of data collection varies from May to June and June to July in other sections.

3. the results report the information not in line with age 15 years participants. The experiences related by the participants are not those of a 15 year old male person and later it was mentioned to be of a 23-67 years.

4. concord: the agreement of the verb and the noun is an issue: data was instead of data were. Language editing might help for this study.

Reviewer #2: This is an interesting study. However, there are a few minor issues that need to be addressed.

1. The study sample needs to be described more elaborately. How many participants were fathers and how many were not? This has important implications on transferability of the findings since the experiences of fathers may be different from that of those who are not fathers.

2. One of the findings of the study was that: "Participants highlighted that socio-cultural norms on gender roles deterred men from actively engaging in childcare and MNCH services." Cultures are not homogenous. They tend to be influenced by ethnicities. South Africa is a very diverse society. For that reason it is even more important to describe the research sample more completely. Were there any Afrikaner men in the sample? Were there non-black men in the sample? The cultures of white male South Africans may differ from those of black males. In the same vein the barriers that men face may differ depending on race. A more elaborate description of the demographic composition of participants is important in contextualizing the research.

3. The limitation of the study is described as follows: "The study was conducted with both men engaged and not engaged with HIV services in Johannesburg district, so may not be generalizable." Since this study is purely qualitative it is more appropriate to speak of transferability rather than generalizability. Generalizability is a concept that is associated with quantitative studies which put premium on the statistical aspects of the studies.

6. PLOS authors have the option to publish the peer review history of their article (what does this mean?). If published, this will include your full peer review and any attached files.

Reviewer #1: **Yes: **Professor Thuledi Makua

Reviewer #2: No

---

## [Author Response · Author response to Decision Letter 0]

4 Dec 2023

EDITOR COMMENTS

1. In your Study Setting section, you mentioned that the study was conducted in South Africa from June-July 2021. In the section that follows, you mentioned that data was collected between May and June 2021. It is not clear why you have such inconsistency in your reporting of dates. 

We have corrected the timelines inconsistencies in the abstract and body. This was a typo. The revised section is provided below (page 6, line 126):

The study was conducted in Johannesburg District, Gauteng province, South Africa from May-July 2021. Johannesburg has a population of approximately 4, 8 million (males 2 419 151; females 2 384 111) (27). 

2. The inclusion and exclusion criteria for the FGD participants in not clearly spelt out. Please provide a clear inclusion criteria that was used to recruit participants. 

We have included the inclusion and exclusion criteria in the “Study design and participants” section (page 8, line 159-166). We have given details on how participants were recruited. Below is the revised section.

Eligible participants were men who were above the age of 18 years old regardless of them having biological or non-biological children, their nationality, ethnicity, race, religion, social status, and educational level. The exclusion criteria were ages less than 18 years and inability to speak in English or local languages. Convenience sampling was used to recruit participants in the study. Male peer counsellors were recruited in the study through an invitation circulated to them by the Health4Men programme lead. Men from local communities were recruited through male peer counselors who mobilized the men to participate in the study during their community mobilization activities.

3. How many FGDs were held? 

We have revised the “Study design and participants” section (page 7, line 148-156) to highlight the number of FGDs conducted and details on the FGDs composition. The section has been revised as follows:

A qualitative exploratory study using FGDs was undertaken. We conducted three (3) FGDs, with a total of 33 men selected from various geographical regions of Johannesburg. One group was composed of male peer counsellors working for Anova Health Institute in the Health4Men programme. The male peer counsellors are based at clinics spread across Johannesburg District and have the responsibility of community mobilization and providing supportive client-centered services at the clinics to men living with HIV. Two groups were constituted of men drawn from local communities. Some of the men had participated in community Health4Men Programme interventions while some had never participated in the interventions. The FGDs comprised of 6-12 participants. The FGDs lasted between 90-180 minutes.

4. In the data analysis section, you mentioned that the NVivo 12 software was used for "coding, categorization, and identification of themes". Further down in the same section, you also mentioned "Each member of the first team independently generated initial codes through open coding." Given that two independent individuals had done this exercise separately, what was the reasoning behind having the second team to re-do the same exercise including re-naming, re-coding, and re-categorizing?

A total of three team members were involved in coding. Two members had been involved in the data collection process. They were both involved in coding to bring individual perspectives which could be linked to their earlier engagement in data collection. The third team member had not been involved in data collection. The third team member reviewed the coding of the merged file from the first two members. The third team member was included to provide a different perspective as they were not part of the data collection process unlike the other two team members. We have provided an explanation in the “Data analysis” section (page 10, lines 196-203; 206-213) on the reason for involving a third team member in coding. We have also paraphrased the content for clarity. The revised sentences are provided below: 

Two research facilitators fluent in local languages transcribed the data, and the generated transcripts were independently reviewed line by line before coding by the team lead who supervised the data collection process. Data were analyzed inductively by organizing it into categories based on themes, concepts, or similar features. A total of 3 team members divided into 2 independent teams analyzed the data. The first team was composed of team members who had been involved in data collection. Each member of these team members independently generated initial codes through open coding. Each member independently coded to bring individual perspectives which could be linked to their earlier engagement in data collection.

The first two team members then merged their files and re-examined the resulting codes and categories to develop final codes, categories, and themes. The re-examination allowed for a discussion on the rationale of the codes, categories, and themes created by each member.

The second team was composed of only one team member who independently reviewed the codes, categories, and themes. The team member had not been involved in data collection. The second team member reviewed the coding of the merged file from the first two members. The third team member was included to provide a different perspective as they were not part of the data collection process unlike the first two team members. 

5. By definition, a theme is a central unifying idea or the bigger issue that emerges from your data. In the results section, you mentioned that fatherhood was one of the key themes identified. How does fatherhood emerge as a theme when in fact it is one of the characteristics used for inclusion into the study? Check the last line in the Results section. How were the themes identified?

Although fatherhood was a key aspect under investigation, it was also considered a construct which needed to be disintegrated into pieces or sub-themes to get a holistic understanding. We have revised the “Data analysis” section (page 11, lines 222-227) to give more detailed description on how themes were identified. Below is the revised section.

The final themes were created through interrogating similarities, differences and connections of created categories. A matching and comparison examination was done on the created categories. Careful consideration was made to ensure that the examination does not outstrip the data of its context. As we matched and compared the categories, we noted emerging patterns and underlying themes through construction of the semantic and latent meanings in the categories. These themes were then used to construct the final narrative.

6. Please make sure to follow all PLOS ONE guidelines when submitting your paper. For example, you did not include line numbers. 

We have include line numbers. 

REVIEWER 1 COMMENTS

1. Is the manuscript presented in an intelligible fashion and written in standard English?

Reviewer #1: No 

We have done a spell check on the entire manuscript and revised accordingly. 

2. Introduction: 

Paragraph 3: first, second and third lines say repeated message in the different ways.

We agree with the reviewer and have revised the sentences by combining them. The revised section is provided below:

Men play a crucial role in MNCH as their influence on decision making and provision of material support can impact their partner’s health care seeking behavior and access to services (8,9).

3. Paragraph 4: third sentence too long loses meaning on the way.

We agree with the reviewer, we have revised the sentence by shortening it. The sentence has been rewritten as follows: 

In the context of HIV/AIDS, evidence indicates that male partners influence their partners’ utilization of HIV prevention services and adherence to antiretroviral treatment (10–13).

4. Methods

Study setting

Paragraph 1: first sentence does not concur with what is in the abstract. When was this study conducted?

We have corrected the timelines inconsistencies in the abstract and body. This was a typo. The revised sentence is provided below:

The study was conducted in Johannesburg District, Gauteng province, South Africa from May-July 2021

5. Second sentence: sounds incomplete.. “aged between 15+ years are estimated”

Whole paragraph 1: fatherhood and age “15+ years” sound incompatible??

The sentences on population and HIV prevalence estimates among men were provided to give the context of HIV among men in Johannesburg and were not the age groups of participants in the study. We have removed the sentences as we believe their inclusion might be misinterpreted. The paragraph has been revised as follows: 

The study was conducted in Johannesburg District, Gauteng province, South Africa from May-July 2021. Johannesburg has a population of approximately 4, 8 million (males 2 419 151; females 2 384 111) (27). This population is spread over 7 regions, with distinct socio-economic characteristics. FGDs were held in various community sites including a shebeen (informal licensed drinking place in the community) and clinic meeting rooms. The sites were selected to ensure representation of sub-district management clusters and recruitment of diverse group of men. The sites were decided upon in consultation between participants and male peer counsellors. The sites were sometimes used by male peer counsellors to conduct community dialogues, educational awareness sessions and during community mobilization campaigns.

6. Study design and participants

Paragraph 1: fifth sentence: when was the study conducted? “May and June 2021” there seems to be no consistency.

We have corrected the timelines inconsistencies in the abstract and body. This was a typo. The revised sentence is provided below:

The study was conducted in Johannesburg District, Gauteng province, South Africa from May-July 2021.

7. Sentence 8: does not align with the age 15 years participants.

The sentences on population and HIV prevalence estimates of men aged 15-49+ were provided to give the context of HIV among men in Johannesburg and were not the age groups of participants in the study. We have removed the sentences as we believe their inclusion might be misinterpreted.

8. Sentence 9: does not align with the age 15 years participants.

The sentences on population and HIV prevalence estimates of men aged 15-49+ were provided to give the context of HIV among men in Johannesburg and were not the age groups of participants in the study. We have removed the sentences as we believe their inclusion might be misinterpreted. 

9. Ethical approval 

Paragraph 1: “Written informed consent was obtained from all participants.” 

Participants underage (aged 15 under methodology above) cannot give consent.

Participants below the ages of 18 years were excluded in the study. All participants were aged above the age of 18 years and could provide informed consent. The sentences on population and HIV prevalence estimates of men aged 15-49+ were provided to give the context of HIV among men in Johannesburg and were not the age groups of participants in the study.

10. Data analysis

Paragraph 1: sentences 5&6: Concord, revise.

We have revised the sentences to harmonize the grammar. The sentences have been revised as follows: 

We analyzed data inductively, organizing segments into categories based on themes, concepts, or similar features. Data were analyzed by two independent teams.

11. Level of involvement

Concord: Double check whether the noun is singular or plural

We have corrected the nouns accordingly. The sentences now read as below:

All participants stressed that poor relations between a father and the children’s mother was a key contributing factor to poor involvement with their children. 

Strained relationships with in-laws and failure to fulfill cultural obligations such as payment of lobola (bride wealth) and/or “damages” (payment for fathering a child without being married to the mother) were reported as other key factors impacting the level of fathers’ involvement in a child’s life.

Culture and gender norms 

Concord: Double check whether the noun is singular or plural

We have corrected the noun from singular to plural. The sentence now reads as below:

Participants in all groups explained that nurturing and caring for children were considered “women’s work” and men were directly or indirectly discouraged to take an active role in such affairs.

12. Results

Paragraph 2: ages of 15+ and The age range of the participants was 23-67 years.’ Please check consistency.

Having revisited our data, the correct age range of participants was 25-74 years. The age range 15-49+ years mentioned under the “Study setting” section was stated to give the population and HIV prevalence estimates of men in this age group living in Johannesburg. The age range 15-49+ years was not that of participants in the study. The statements stating the population and HIV prevalence estimates of men age range 15-49+ years have been removed as we believe their inclusion might be misinterpreted. 

REVIEWER 1 FINAL COMMENTS 

Very important and legal issue of participants’ age to take part in the study has been overlooked when the study involved age 15+ participants. Towards the end the study changes from age 15+ to 23-67 years. There was no mention of steps taken to with regard to ages under 18. For minors younger than 18 years of age to participate in a research study, parental or guardian permission must be obtained. For minors a youth assent form is required.

As mentioned above, the age range 15-49+ years which was stated under the “Study setting” section was included to give the population and HIV prevalence estimates of men in this age group living in Johannesburg. The age range 15-49+ years was not that of participants in the study. All participants in the study were above 18 years as this was one of the inclusion criterion. The statements stating the population and HIV prevalence estimates of men age range 15-49+ years have been removed as we believe their inclusion might be misinterpreted. 

1. The period of data collection varies from May to June and June to July in other sections.

We agree with the reviewer that there were discrepancies in timelines on data collection. This was a typo. We have corrected the timelines inconsistencies in the abstract and body. The revised sentence is provided below:

The study was conducted in Johannesburg District, Gauteng province, South Africa from May-July 2021.

2. The results report the information not in line with age 15 years participants. The experiences related by the participants are not those of a 15 year old male person and later it was mentioned to be of a 23-67 years.

We revisited our data and the correct age range of the participants is 25-74 years. The age range 15-49+ years mentioned under the “Study setting” section was not that of participants in the study. The 15-49+ years age range represented the age range of the population and HIV prevalence estimates of men in living in Johannesburg. The statements stating the population and HIV prevalence estimates of men age range 15-49+ years have been removed as we believe their inclusion might be misinterpreted. 

3. Concord: the agreement of the verb and the noun is an issue: data was instead of data were. Language editing might help for this study.

We have edited the whole document and addressed all grammatical errors highlighted by the reviewers. In this instance, the sentence (page, lines 197-198) referred to by the reviewer has been revised as follows: 

Data were analyzed inductively by organizing it into categories based on themes, concepts, or similar features.

REVIEWER 2 COMMENTS 

1. The study sample needs to be described more elaborately. How many participants were fathers and how many were not? This has important implications on transferability of the findings since the experiences of fathers may be different from that of those who are not fathers. 

We agree with the reviewer and have included a table with the demographic characteristics of the participants and description of the participants in the “Results “section. The table with demographic characteristics of participants is provided below: 

Table 1: Profile of the participants 

Variable n

Age group 

25-34 7

35-44 12

45-54 10

55-64 1

65-74 1

Missing responses 2

Employment status 

Employed 22

Unemployed 9

Missing responses 2

Race 

Black African 31

Missing responses 2

Number of biological children 

0 1

1-2 12

3-4 12

5-6 4

7-8 2

Missing responses 2

Number of non-biological children 

0 9

1-2 16

3-4 5

5-6 1

Missing responses 2

Considers himself to be involved father 

Yes 31

No 0

Missing responses 2

Currently resides with any child 

Yes 24

No 7

Missing responses 2

2. One of the findings of the study was that: "Participants highlighted that socio-cultural norms on gender roles deterred men from actively engaging in childcare and MNCH services." Cultures are not homogenous. They tend to be influenced by ethnicities. South Africa is a very diverse society. For that reason it is even more important to describe the research sample more completely. Were there any Afrikaner men in the sample? Were there non-black men in the sample? The cultures of white male South Africans may differ from those of black males. In the same vein the barriers that men face may differ depending on race. A more elaborate description of the demographic composition of participants is important in contextualizing the research. 

We agree with the reviewer and have included a description of the participant’ characteristics in the “Results” section, paragraph three. The description is provided below: 

The ages of the participants ranged from 25-74 years (see Table 1). The majority of participants (12) were in the age group 35-44 years. The majority of participants (22) were employed while nine (9) of the participants were unemployed. All the participants identified as Black African. Twelve (12) of the participants reported that they had 1-2 biological children, another 12 participants reported that they had 2-4 biological children and only one participant reported not having a biological children. The majority of participants (16) reported having non-biological children. All participants (31) considered themselves to be involved fathers. The majority of participants (24) reported residing with their children. Participants were of mixed ethnic backgrounds, including Zulu, Xhosa, Sepedi, Venda and Tsonga. 

3. The limitation of the study is described as follows: "The study was conducted with both men engaged and not engaged with HIV services in Johannesburg district, so may not be generalizable." Since this study is purely qualitative it is more appropriate to speak of transferability rather than generalizability. Generalizability is a concept that is associated with quantitative studies which put premium on the statistical aspects of the studies. 

We agree with the reviewer and have revised the statement in the “Limitations” section to state as follows: 

The transferability of the study findings may be limited to men who are not interacting with health or community services. However, the findings offer valuable insights for strengthening MNCH services and integration with HIV services.

---

## [Editor Report · Decision Letter 1]

26 Dec 2023

Using Fatherhood to engage men in HIV services via Maternal, Neonatal and Child Health entry points in South Africa

PONE-D-23-16181R1

Dear Dr. Chinyandura,

We’re pleased to inform you that your manuscript has been judged scientifically suitable for publication and will be formally accepted for publication once it meets all outstanding technical requirements.

Kind regards,

Edward Chiyaka, Ph.D., MSc

Academic Editor

PLOS ONE

---

## [Editor Report · Acceptance letter]

12 Mar 2024

PONE-D-23-16181R1 

PLOS ONE

Dear Dr. Chinyandura, 

I'm pleased to inform you that your manuscript has been deemed suitable for publication in PLOS ONE. Congratulations! Your manuscript is now being handed over to our production team.

Kind regards, 

on behalf of

Dr. Edward Chiyaka 

Academic Editor

PLOS ONE